# Prediction of Strain Path Changing Effect on Forming Limits of AA 6111-T4 Based on a Shear Ductile Fracture Criterion

**Silin Luo [1], Gang Yang [1], Yanshan Lou [2,*] and Yongqian Xu [3]**

[1]  School of Mechatronics and Mold Engineering, Taizhou Vocational College of Science & Technology, Taizhou 318020, China; Lsilin@tzvcst.edu.cn (S.L.); GangYang@tzvcst.edu.cn (G.Y.)
[2]  School of Mechanical Engineering, Xi'an Jiaotong University, 28 Xianning West Road, Xi'an 710049, China
[3]  State Key Laboratory of High-Performance Complex Manufacturing, Central South University, Changsha 410083, China; yongqian.xu@csu.edu.cn
*  Correspondence: ys.lou@xjtu.edu.cn; Tel.: +86-186-9183-6771

**Abstract:** Strain path changing is a phenomenon in the stamping of complex panels or multiple-step stamping processes. In this study, the influence of the strain path changing effect was investigated and assessed for an aluminum alloy of 6111-T4 with a shear ductile fracture criterion. Plastic deformation of the alloy was modeled by an anisotropic Drucker yield function with the assumption of normal anisotropy. Then the shear ductile fracture criterion was calibrated by the fracture strains at uniaxial tension, plane strain tension and equibiaxial tension under proportional loading conditions. The calibrated fracture criterion was utilized to predict forming limit curves (FLCs) of the alloy stretched under bilinear strain paths. The analyzed bilinear strain paths included biaxial tension after uniaxial tension, plane strain tension and equibiaxial tension. The predicted FLCs of bilinear strain paths were compared with experimental results. The comparison showed that the shear ductile fracture criterion could reasonably describe the effect of strain path changing on FLCs, but its accuracy was poor for some bilinear paths, such as uniaxial tension followed by equibiaxial tension and equibiaxial tension followed by plane strain tension. Kinematic hardening is suggested to substitute the isotropic hardening assumption for better prediction of FLCs with strain path changing effect.

**Keywords:** shear ductile fracture; forming limit curve; strain path changing; sheet metal forming



## 1. Introduction

The accurate prediction of failure during sheet metal forming is a big challenge in the numerical design of sheet metal forming processes. To describe the largest strain that sheet metals can plastically deform without failure during stamping, Goodwin [1] introduced the idea of forming limit curves (FLCs), which set the maximum plastic deformation in the space of the major and minor strains. Since its proposal, FLC has been widely used to define the forming limit of sheet metal forming. An FLC is generally measured by experiments, which are designed for linear strain paths. However, the sheet metal forming processes of complex panels and multiple-step forming involves dramatic strain path changing during stamping. Experimental studies show that strain path changing strongly influences the formability of sheet metals. Graf and Hosford [2] conducted experiments of AA2008-T4 to investigate the effect of strain path changing on FLCs. One year later, Graf and Hosford [3] shared their experimental results on the influence of strain path changing on FLCs of AA6111-T4. Their experiments have shown that the experimental FLCs under proportional strain path are proper for the failure prediction of sheet metal forming with approximately linear strain paths, but the FLC is somewhat wrongly used for stamping of complex-shaped panels or multiple-step forming with strong strain path changing.

Theoretically, Hill [4] proposed a localized necking model for the failure modeling of sheet metal forming. Meanwhile, Swift [5] introduced a diffuse necking model. Generally, Hill's localized necking model is used to predict the left-hand side of an FLC, while Swift's

diffuse necking model is used to predict the right-hand side FLC. Therefore, these two models are generally combined together and referred to as the Hill-Swift model. Marciniak and Kuczynski [6] proposed an imperfection-based model to predict FLCs of sheet metals. The Marciniak-Kuczynski model is widely used and analyzed since its proposal. The Marciniak-Kuczynski model can be used to take account of the effect of yield functions, strain rate effect and even microstructures in FLC prediction [7]. Cao et al. [8] applied the Marciniak-Kuczynski model to investigate the effect of strain path changing on FLCs. Stoughton [9] showed that the forming limit for both proportional and nonproportional loading could be explained from a single criterion, which is based on the state of stress rather than the state of strain, thereby introducing the definition of forming a limit stress diagram. Based on the crystal plasticity theory in conjunction with the Marciniak-Kuczynski model, Wu et al. [10] suggested that the forming limit stress diagram is much more favorable than the forming limit strain diagram in representing forming limits in the numerical simulation of sheet metal forming processes. Alternatively, Stoughton and Yoon [11] proposed a new type of forming limit diagrams based on a polar representation of the effective plastic strain and showed that the new diagram is an effective solution to the problem of nonlinear effects. Korkolis and Kyriakides [12] studied the effect of strain path changing effect on the failure of inflated aluminum tubes and concluded that the amount of plastic pre-strain also affects the failure stress and strain during tube hydroforming.

Since the beginning of the 20th century, ductile fracture has attracted increasing efforts for failure prediction in sheet metal forming as the wide employment of advanced high strength steels and aluminum alloys. It is because the lightweight metals fail mainly by a ductile fracture with little necking [13,14]. Various ductile fracture criteria were proposed for the modeling of ductile fracture under different loading conditions for sheet metals. Bai and Wierzbicki [15] modified the Mohr-Coulomb criterion for ductile fracture modeling of advanced high strength steels. Mohr and Marcedat [16] proposed a phenomenological Hosford-Coulomb model for the prediction of ductile fracture at low-stress triaxiality. Ductile fracture is observed to take place along the direction of the maximum shear stress in different stress states of compression, shear and tension [17]. Based on this experimental observation, Lou et al. [18] proposed a shear ductile fracture criterion based on the micro-mechanism of ductile fracture for the nucleation, growth and coalescence of voids. Thereafter, two modifications were developed to consider a changeable cutoff value for the stress triaxiality [19] and to describe fracture in shear, uniaxial tension, plane strain tension and equibiaxial tension of sheet metals [20]. Hu et al. [21] and Mu et al. [22] developed two different fracture models considering the effect of the maximum shear stress. Sun et al. [23] proposed a new method by directly utilizing original measured data of the stress-strain relation in the Marciniak-Kuczynski model to predict the FLC of an aluminum alloy sheet. Cao et al. [24] developed a fracture model coupled with the Johnson-Cook plasticity model to investigate the strain rate effect for 7050-T7451 aluminum alloy.

Though ductile fracture criteria are increasingly used in the fracture prediction of sheet metal forming, the effect of strain path changing is still not clear. In this study, the strain path changing effect is investigated by using a recently proposed ductile fracture criterion [18], which is referred to as DF2012. The DF2012 criterion is first calibrated for AA6111-T4 under proportional loading. Then the calibrated criterion is used to describe the pre-strain effect of uniaxial tension, plane strain tension and equibiaxial tension on the fracture-forming limit curve (FFLC). The predicted FFLCs under bilinear strain paths are compared with experimental results to analyze its predictability of the pre-strain influence on FFLCs. Reasons are discussed for the improper prediction of FFLCs under some strain path pairs, such as uniaxial tension followed by equibiaxial tension and equibiaxial tension followed by plane strain tension.

## 2. Modeling of Plastic Deformation

The material used in this study was an aluminum alloy sheet of AA6111-T4. Its thickness was 1.05 mm. The chemical composition of the alloy was approximately of 0.85% Si, 0.75% Cu, 0.65% Mg, 0.25% Fe, 0.22% Mn and 0.03% Cr. The grain of the alloy was pancake-shaped. Experiments were conducted using dog bone specimens along three directions of 0°, 45° and 90°. Strain hardening behavior was described by the Hollomon law as $\sigma = 563\varepsilon^{0.255}$ MPa. The anisotropic Lankford values or R-values were also measured along the rolling, diagonal and transverse directions as 0.67, 0.63 and 0.78, respectively. The average R-value was calculated as $\bar{r} = (r_0 + 2r_{45} + r_{90})/4 = 0.68$, and $\Delta r$ was calculated as $\Delta r = (r_0 - 2r_{45} + r_{90})/2 = 0.1$. $\Delta r$ was comparatively small compared with $\bar{r}$. Accordingly, normal anisotropy could be assumed for simplicity purposes. All the experiments in this paragraph and the FFLC tests under proportional and nonproportional loading paths below were conducted by Graf and Hosford [3]. Readers are strongly suggested to refer to the original paper by Graf and Hosford [3] for the details of the experimental procedures and results.

FFLCs under proportional loading were measured by the standard punch stretching tests for AA6111-T4 sheets. The strains were measured by the distortion of circles for, which its diameter was 2.54 mm before stretching. The measured fracture strain was about 0.35 at uniaxial tension, 0.17 under plane strain tension and 0.244 at equibiaxial tension. The sheets were also prestrained in uniaxial tension, plane strain tension and biaxial tension to different strain levels. Then specimens were cut from the prestrained sheets and stretched further by punch stretching to measure the FFLC for each combination of prestrain strain paths and levels. The measured FFLC under bilinear strain paths was used for the evaluation of the analytical prediction of fracture limits. Details of FFLC experiments were suggested to refer to Graf and Hosford [3].

For the sake of simplicity, the material is assumed to be normal anisotropic. There are many anisotropic yield functions proposed for both sheet and bulk metals [25–32]. The anisotropic Drucker function [33] is used to describe the normal anisotropy of the metal in this study. Similar to the non-quadratic yield functions, the anisotropic Drucker function can differentiate the difference in yielding between face-centered cubic (FCC) and body-centered cubic (BCC) metals. The anisotropic Drucker function is formulated as:

$$f(\sigma_{ij}) = \left(J_2'^3 - c J_3'^2\right)^{1/6} = \overline{\sigma} \tag{1}$$

In the anisotropic Drucker function above, $J_2'$ and $J_3'$ are referred to as the second and third invariants of $\mathbf{s}'$, which is computed as follows:

$$J_2' = \frac{1}{2}\mathbf{s}' : \mathbf{s}' = -s_{11}'s_{22}' - s_{22}'s_{33}' - s_{11}'s_{33}' + s_{12}'^2 + s_{23}'^2 + s_{13}'^2 \tag{2}$$

$$J_3' = \det\left(\mathbf{s}'\right) = s_{11}'s_{22}'s_{33}' + 2s_{12}'s_{23}'s_{13}' - s_{11}'s_{23}'^2 - s_{22}'s_{13}'^2 - s_{33}'s_{12}'^2 \tag{3}$$

where $\mathbf{s}'$ is computed as:

$$\mathbf{s}' = \mathbf{L}'\boldsymbol{\sigma} \tag{4}$$

Here $\mathbf{L}'$ is the fourth-order linear transformation given by:

$$\mathbf{L}' = \begin{bmatrix} (c_2' + c_3')/3 & -c_3'/3 & -c_2'/3 & 0 & 0 & 0 \\ -c_3'/3 & (c_3' + c_1')/3 & -c_1'/3 & 0 & 0 & 0 \\ -c_2'/3 & -c_1'/3 & (c_1' + c_2')/3 & 0 & 0 & 0 \\ 0 & 0 & 0 & c_4' & 0 & 0 \\ 0 & 0 & 0 & 0 & c_5' & 0 \\ 0 & 0 & 0 & 0 & 0 & c_6' \end{bmatrix} \tag{5}$$

Then the linearly transformed stress tensor $\mathbf{s}'$ can be explicitly expressed as:

$$
\mathbf{s}' = \mathbf{L}' \boldsymbol{\sigma} =
\begin{bmatrix}
(c_2'+c_3')/3 & -c_3'/3 & -c_2'/3 & 0 & 0 & 0 \\
-c_3'/3 & (c_3'+c_1')/3 & -c_1'/3 & 0 & 0 & 0 \\
-c_2'/3 & -c_1'/3 & (c_1'+c_2')/3 & 0 & 0 & 0 \\
0 & 0 & 0 & c_4' & 0 & 0 \\
0 & 0 & 0 & 0 & c_5' & 0 \\
0 & 0 & 0 & 0 & 0 & c_6'
\end{bmatrix}
\begin{bmatrix}
\sigma_{xx} \\ \sigma_{yy} \\ \sigma_{zz} \\ \sigma_{yz} \\ \sigma_{zx} \\ \sigma_{xy}
\end{bmatrix}
$$

$$
=
\begin{bmatrix}
\frac{c_2'+c_3'}{3}\sigma_{xx} - \frac{c_3'}{3}\sigma_{yy} - \frac{c_2'}{3}\sigma_{zz} \\
-\frac{c_3'}{3}\sigma_{xx} + \frac{c_3'+c_1'}{3}\sigma_{yy} - \frac{c_1'}{3}\sigma_{zz} \\
-\frac{c_2'}{3}\sigma_{xx} - \frac{c_1'}{3}\sigma_{yy} + \frac{c_1'+c_2'}{3}\sigma_{zz} \\
c_4'\sigma_{yz} \\
c_5'\sigma_{zx} \\
c_6'\sigma_{xy}
\end{bmatrix}
\tag{6}
$$

For sheet metal forming under plane-stress conditions, the linearly transformed tensor is computed as:

$$
\mathbf{s}' = \mathbf{L}' \boldsymbol{\sigma} =
\begin{bmatrix}
(c_2'+c_3')/3 & -c_3'/3 & -c_2'/3 & 0 & 0 & 0 \\
-c_3'/3 & (c_3'+c_1')/3 & -c_1'/3 & 0 & 0 & 0 \\
-c_2'/3 & -c_1'/3 & (c_1'+c_2')/3 & 0 & 0 & 0 \\
0 & 0 & 0 & c_4' & 0 & 0 \\
0 & 0 & 0 & 0 & c_5' & 0 \\
0 & 0 & 0 & 0 & 0 & c_6'
\end{bmatrix}
\begin{bmatrix}
\sigma_{xx} \\ \sigma_{yy} \\ 0 \\ 0 \\ 0 \\ \sigma_{xy}
\end{bmatrix}
$$

$$
=
\begin{bmatrix}
\frac{c_2'+c_3'}{3}\sigma_{xx} - \frac{c_3'}{3}\sigma_{yy} \\
-\frac{c_3'}{3}\sigma_{xx} + \frac{c_3'+c_1'}{3}\sigma_{yy} \\
-\frac{c_2'}{3}\sigma_{xx} - \frac{c_1'}{3}\sigma_{yy} \\
0 \\
0 \\
c_6'\sigma_{xy}
\end{bmatrix}
\tag{7}
$$

In the anisotropic Drucker function, the effect of the third invariant $J_3'$ is modified by $c$, which is calibrated to be 1.226 and 2.0 for BCC and FCC metals. Moreover, six anisotropic parameters are introduced in the linear transformation tensor of $L'$, among which four are related to anisotropic plastic behavior. When the anisotropic parameters are $c_i' = 1.8365$ ($i = 1 \sim 6$) with $c = 2$, the anisotropic Drucker function reduces to isotropic for FCC metals, while the isotropic Drucker function for BCC metals is recovered when $c_i' = 1.7909$ ($i = 1 \sim 6$) with $c = 1.226$.

The derivatives of the anisotropic Drucker function for plane-stress conditions are calculated as follows:

$$
d\varepsilon_{xx} = \frac{\partial \overline{\sigma}_y}{\partial \sigma_{xx}} = \frac{1}{6}\left(J_2'^3 - c J_3'^2\right)^{-5/6}\left(3J_2'^2 \frac{\partial J_2'}{\partial \sigma_{xx}} - 2c J_3' \frac{\partial J_3'}{\partial \sigma_{xx}}\right)
\tag{8}
$$

$$
d\varepsilon_{yy} = \frac{\partial \overline{\sigma}_y}{\partial \sigma_{yy}} = \frac{1}{6}\left(J_2'^3 - c J_3'^2\right)^{-5/6}\left(3J_2'^2 \frac{\partial J_2'}{\partial \sigma_{yy}} - 2c J_3' \frac{\partial J_3'}{\partial \sigma_{yy}}\right)
\tag{9}
$$

$$
d\varepsilon_{xy} = \frac{\partial \overline{\sigma}_y}{\partial \sigma_{xy}} = \frac{1}{6}\left(J_2'^3 - c J_3'^2\right)^{-5/6}\left(3J_2'^2 \frac{\partial J_2'}{\partial \sigma_{xy}} - 2c J_3' \frac{\partial J_3'}{\partial \sigma_{xy}}\right)
\tag{10}
$$

with $d\varepsilon_{yz} = d\varepsilon_{zx} = 0$ and:

$$
\frac{\partial J_2'}{\partial \sigma_{xx}} = \frac{(c_2'+c_3')s_{11}' - c_3's_{22}' - c_2's_{33}'}{3}
\tag{11}
$$

$$\frac{\partial J'_2}{\partial \sigma_{yy}} = \frac{-c'_3 s'_{11} + (c'_1 + c'_3)s'_{22} - c'_1 s'_{33}}{3} \tag{12}$$

$$\frac{\partial J'_2}{\partial \sigma_{xy}} = 2c'_6 s'_{12} \tag{13}$$

$$\frac{\partial J'_3}{\partial \sigma_{xx}} = \frac{(c'_2 + c'_3)s'_{22}s'_{33} - c'_3 s'_{11}s'_{33} - c'_2 s'_{11}s'_{22} + c'_2 s'^2_{12}}{3} \tag{14}$$

$$\frac{\partial J'_3}{\partial \sigma_{yy}} = \frac{-c'_3 s'_{22}s'_{33} + (c'_1 + c'_3)s'_{11}s'_{33} - c'_1 s'_{11}s'_{22} + c'_1 s'^2_{12}}{3} \tag{15}$$

$$\frac{\partial J'_3}{\partial \sigma_{xy}} = -2c'_6 s'_{12}s'_{33} \tag{16}$$

$s'_{ij}$ is the vector transformed in Equation (7) and calculated in the forms of:

$$s'_{11} = \frac{(c'_2 + c'_3)\sigma_{xx} - c'_3 \sigma_{yy}}{3} \tag{17}$$

$$s'_{22} = \frac{-c'_3 \sigma_{xx} + (c'_1 + c'_3)\sigma_{yy}}{3} \tag{18}$$

$$s'_{33} = \frac{-c'_2 \sigma_{xx} - c'_1 \sigma_{yy}}{3} \tag{19}$$

$$s'_{12} = c'_6 \sigma_{xy}, \; s'_{23} = s'_{13} = 0 \tag{20}$$

Considering the incompressibility of plastic deformation for continuum metals, the thickness strain can be computed as $d\varepsilon_{zz} = -d\varepsilon_{xx} - d\varepsilon_{yy}$.

For AA6111-T4, the anisotropic Drucker function was calibrated by the mean R-values by the assumption that the material was normal anisotropic. The calibrated anisotropic parameters are listed in Table 1. The AA6111-T4 yield surface is described in Figure 1. The anisotropic Drucker yield surface was noted to describe lower strength under plane strain tension than the von Mises function. Furthermore, the predicted R-values and uniaxial tensile yield stress were insensitive to the loading direction since normal anisotropy was assumed for AA6111-T4. The predicted R-value was 0.68, which was identical to the experimental mean R-value.

**Table 1.** Parameters of the anisotropic Drucker function.

| $c$ | $c'_1$ | $c'_2$ |
|---|---|---|
| 2.0000 | 1.9392 | 1.9392 |

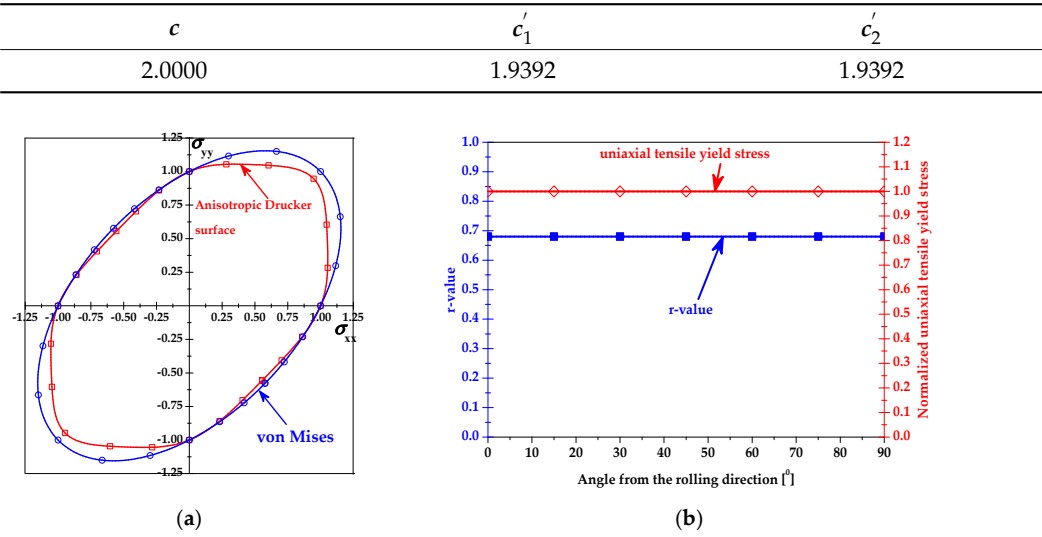

(a)     (b)

**Figure 1.** Anisotropic Drucker yield surface for AA6111-T4: (**a**) yield surface; (**b**) uniaxial tensile yield stresses and R-values.

### 3. Shear Ductile Fracture Criterion

To model fracture in metal forming processes, various ductile fracture criteria [15–22] were developed based on different assumptions. Recently, Lou et al. [18] developed a micromechanism-inspired ductile fracture criterion for nucleation, growth and shear coalescence of voids in the form of:

$$\left(\frac{2\tau_{max}}{\overline{\sigma}}\right)^{C_1}\left(\frac{\langle 1+3\eta\rangle}{2}\right)^{C_2}\overline{\varepsilon}_f^p = C_3 \quad \langle x\rangle = \begin{cases} x & if\, x \geq 0 \\ 0 & if\, x < 0 \end{cases} \tag{21}$$

where $\overline{\sigma}$ represents the equivalent stress, $\tau_{max}$ is the largest shear stress, $\eta$ is the stress triaxiality, and $\overline{\varepsilon}_f^p$ is the equivalent plastic strain at the onset of fracture. $\eta$ is computed as $\eta = \sigma_m/\overline{\sigma}$ with $\sigma_m$ as the mean stress. There are three fracture parameters of $C_1$, $C_2$ and $C_3$, which need to be calibrated by experimental data points. For a sheet metal stretched by two principal stresses along RD and TD, the stress state is assumed to be $(\sigma_{xx}, \sigma_{yy})$ and the stress ratio is defined as $\alpha = \sigma_{yy}/\sigma_{xx}$. Then the strain increment ratio under this loading condition is computed with the assumption of the associate flow rule as below:

$$\beta = \frac{d\varepsilon_{yy}}{d\varepsilon_{xx}} \tag{22}$$

The strain increments of $d\varepsilon_{xx}$ and $d\varepsilon_{yy}$ are given in Equations (8) and (9), respectively. Moreover, $\phi$ is defined as the ratio of the stress $\sigma_{xx}$ to the anisotropic Drucker effective stress computed in Equation (1) as below:

$$\phi = \frac{\sigma_{xx}}{\overline{\sigma}} \tag{23}$$

Under biaxial tension with $\sigma_{xx} \geq \sigma_{yy} \geq 0$, the normalized maximum shear stress is obtained as:

$$\frac{\tau_{max}}{\overline{\sigma}} = \frac{\sigma_{xx}}{2\overline{\sigma}} = \frac{\phi}{2} \tag{24}$$

The stress triaxiality is calculated as:

$$\eta = \frac{\sigma_m}{\overline{\sigma}} = \frac{\sigma_{xx}+\sigma_{yy}}{3\overline{\sigma}} = \frac{\phi(1+\alpha)}{3} \tag{25}$$

The equivalent plastic strain increment can be calculated with the equation below:

$$\overline{\sigma}d\overline{\varepsilon} = \sigma_{xx}d\varepsilon_{xx} + \sigma_{yy}d\varepsilon_{yy} = \sigma_{xx}d\varepsilon_{xx}(1+\alpha\beta) \tag{26}$$

Then the equivalent plastic strain increment is obtained as:

$$d\overline{\varepsilon} = \frac{\sigma_{xx}d\varepsilon_{xx}(1+\alpha\beta)}{\overline{\sigma}} = \phi(1+\alpha\beta)d\varepsilon_{xx} \tag{27}$$

With Equations (24)–(27), the DF2012 criterion under proportional biaxial tension can be reformulated as:

$$\phi^{C_1}\left(\frac{\langle 1+\phi(1+\alpha)\rangle}{2}\right)^{C_2}\phi(1+\alpha\beta)\varepsilon_{xx}^f = C_3 \tag{28}$$

The above equation can be expressed in the form below:

$$\phi^{C_1}\left(\frac{\langle 1+\phi(1+\alpha)\rangle}{2}\right)^{C_2} = \frac{C_3}{\phi(1+\alpha\beta)\varepsilon_{xx}^f} \tag{29}$$

Logarithmic operation is applied on both sides of the above equation, which gives:

$$C_1 \log \phi + C_2 \log \left( \frac{\langle 1 + \phi(1+\alpha) \rangle}{2} \right) = \log \left( \frac{C_3}{\phi(1+\alpha\beta)\varepsilon_{xx}^f} \right) \tag{30}$$

For AA6111-T4, the relations between $\alpha$, $\phi$ and $\beta$ are obtained with the anisotropic Drucker function, as illustrated in Figure 2.

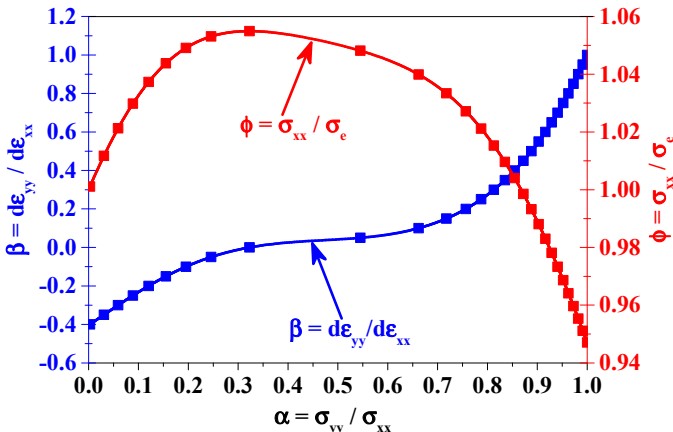

**Figure 2.** Relations between $\alpha$, $\phi$ and $\beta$ for AA6111-T4 modeled by the anisotropic Drucker function.

For uniaxial tension, $\alpha = 0$, $\phi = 1$, and $\beta = -r/(1+r) \approx -0.404$ and then $C_3 = \varepsilon_{xx[UT]}^f = \bar{\varepsilon}_{[UT]}^f$, where $\varepsilon_{xx[UT]}^f$ and $\bar{\varepsilon}_{[UT]}^f$ are the uniaxial tensile fracture strain components along RD and the equivalent plastic strain to fracture of uniaxial tension.

For plane strain tension, $\alpha = \alpha_{[PS]}$, $\phi = \phi_{[PS]}$, and $\beta = 0$. Then the DF2012 criterion in Equation (30) reduces to:

$$C_1 \log \phi_{[PS]} + C_2 \log \left( \frac{\left\langle 1 + \phi_{[PS]}\left(1+\alpha_{[PS]}\right) \right\rangle}{2} \right) = \log \left( \frac{C_3}{\phi_{[PS]}\varepsilon_{xx[PS]}^f} \right) \tag{31}$$

with $\varepsilon_{xx[PS]}^f$ as the fracture strain component along RD under plane strain tension along RD.

For the equibiaxial tension with $d\varepsilon_{xx} = d\varepsilon_{yy}$, $\alpha = \alpha_{[EB]}$, $\phi = \phi_{[EB]}$, and $\beta = 1$. Then the DF2012 criterion in Equation (30) reduces to:

$$C_1 \log \phi_{[EB]} + C_2 \log \left( \frac{\left\langle 1 + \phi_{[EB]}\left(1+\alpha_{[EB]}\right) \right\rangle}{2} \right) = \log \left( \frac{C_3}{\phi_{[EB]}\left(1+\alpha_{[EB]}\right)\varepsilon_{xx[EB]}^f} \right) \tag{32}$$

with $\varepsilon_{xx[EB]}^f$ as the fracture strain component along RD under equibiaxial tension.

Since $C_3 = \varepsilon_{xx[UT]}^f = \bar{\varepsilon}_{[UT]}^f$, $C_1$ and $C_2$ are solved with Equations (31) and (32) as follows:

$$\begin{bmatrix} C_1 \\ C_2 \end{bmatrix} = \begin{bmatrix} \log \phi_{[PS]} & \log \left( \frac{\langle 1+\phi_{[PS]}(1+\alpha_{[PS]}) \rangle}{2} \right) \\ \log \phi_{[EB]} & \log \left( \frac{\langle 1+\phi_{[EB]}(1+\alpha_{[EB]}) \rangle}{2} \right) \end{bmatrix}^{-1} \begin{bmatrix} \log \left( \frac{C_3}{\phi_{[PS]}\varepsilon_{xx[PS]}^f} \right) \\ \log \left( \frac{C_3}{\phi_{[EB]}(1+\alpha_{[EB]})\varepsilon_{xx[EB]}^f} \right) \end{bmatrix} \tag{33}$$

## 4. Modeling of Strain Path Changing Effect on FFLC

Graf and Hosford [3] conducted forming limit tests for AA6111-T4 from uniaxial tension to equibiaxial tension. The fracture strains were measured for the alloy. Table 2 summarizes the measured fracture strain pairs under uniaxial tension, plane strain tension and equibiaxial tension. With the analysis in Section 3 and Equation (33), the fracture parameters of the DF2012 criterion can be easily calculated as $C_1 = 8.8484$, $C_2 = 0.0417$ and $C_3 = 0.2900$. For the computation of fracture strain under bilinear strain path, we first assume that the pre-strain strain path is $\beta_1$, the corresponding stress ratio $\alpha_1$ and the magnitude of the pre-strain $\varepsilon_{xx(1)}$. Then the following strain path, stress ratio and strain increment up to fracture are denoted as $\beta_2, \alpha_2$ and $\varepsilon_{xx(2)}$. According to Equation (28), it is easy to obtain the secondary strain up to fracture by solving Equation (34) as below:

$$\phi_1^{C_1}\left(\frac{\langle 1 + \phi_1(1 + \alpha_1)\rangle}{2}\right)^{C_2}\phi_1(1 + \alpha_1\beta_1)\varepsilon_{xx(1)} + \phi_2^{C_1}\left(\frac{\langle 1 + \phi_2(1 + \alpha_2)\rangle}{2}\right)^{C_2}\phi_2(1 + \alpha_2\beta_2)\varepsilon_{xx(2)}^f = C_3 \tag{34}$$

**Table 2.** Fracture strains for AA6111-T4.

| Loading Conditions | Uniaxial Tension | Plane Strain Tension |
|---|---|---|
| Maximum principal strain | 0.290 | 0.170 |
| Minimum principal strain | 0.117 | 0 |

Then the fracture limit strain with a pre-strain of $\varepsilon_{xx(1)}$ and strain path of $\beta_1$ followed by the strain path of $\beta_2$ is obtained as:

$$\begin{bmatrix} \varepsilon_{xx} \\ \varepsilon_{yy} \end{bmatrix} = \begin{bmatrix} \varepsilon_{xx(1)} + \varepsilon_{xx(2)}^f \\ \varepsilon_{yy(1)} + \varepsilon_{yy(2)}^f \end{bmatrix} = \begin{bmatrix} \varepsilon_{xx(1)} + \varepsilon_{xx(2)}^f \\ \beta_1\varepsilon_{xx(1)} + \beta_2\varepsilon_{xx(2)}^f \end{bmatrix} \tag{35}$$

Then the FFLC under bilinear strain paths are easily predicted by keeping $\beta_1$ as uniaxial tension, plane strain tension, equibiaxial tension or other strain paths and setting $\varepsilon_{xx(1)}$ as different values prestrain, and then varying the strain path of secondary plastic deformation of $\beta_2$ from uniaxial tension to equibiaxial tension.

The FFLC was predicted by the calibrated DF2012 criterion as compared with experimental measurement in Figure 3. It was obvious that the DF2012 criterion predicts the FFLC of AA6111-T4 accurately from uniaxial tension to equibiaxial tension. The difference between prediction and experimental measurement was acceptable for the failure modeling of sheet metal.

The effect of pre-strain under uniaxial tension was theoretically predicted by the DF2012 criterion and compared with experimental results in Figure 4. From both prediction and experiments, pre-strain under uniaxial tension had little impact on the fracture strain between uniaxial tension and plane strain tension. On the other hand, the pre-strain under uniaxial tension tremendously improves the formability when the following strain path was from plane strain tension to balanced biaxial tension. The improvement in fracture strain reached the maximum when the strain path was first uniaxial tension followed by balanced biaxial tension. This observation indicated that forming processed and tools could be designed to reach better drawability of metal sheets when the strain path follows the sequence of uniaxial tension and equibiaxial tension. Moreover, it was noted that the fracture strain under uniaxial tension followed by uniaxial tension was about 0.06 higher than that of solely uniaxial tension. This was because the unloading improves the fracture strain for uniaxial tension, unloading and reloaded under uniaxial tension. Finally, the experimental measured FFLC with the second strain path from plane strain tension to balanced biaxial tension was higher than the FFLC from prediction. The prediction error was due to the simple isotropic hardening was assumed in this study. The unloading effect was expected to be the reason for higher fracture strain for these bilinear strain paths, but it was neglected in the plasticity modeling. Analytically, kinematic hardening or anisotropic

hardening were suggested to be employed for better predicting accuracy of FFLCs for strain path changing effect.

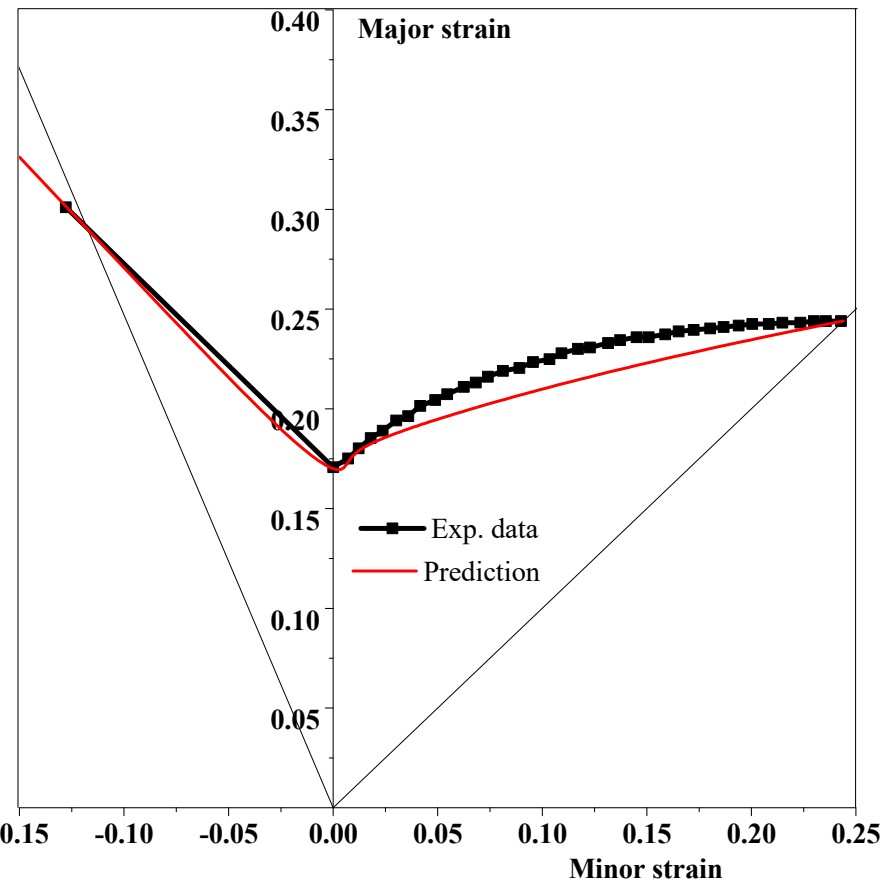

**Figure 3.** Fracture-forming limit curve FFLC) comparison between experiments and prediction by the DF2012 criterion under proportional loading.

The predicted effect of the pre-strain under plane strain tension was compared with experimental results in Figure 5. First, it should be noted that there was a quite large difference in experimental fracture strain between monotonic loading and loading-unloading-reloading even with the identical strain path. Taking the plane strain as an example, 0.05 prestrain under plane strain tension improved about 9.4% fracture strain under plane strain tension than monotonic plane strain tension, as observed in Figure 5. 0.11 prestrain under plane strain tension raised the fracture strain by about 26.2% compared with the fracture strain under monotonically plane strain tension. This could not be explained theoretically here with isotropic hardening. The evolution of yield surfaces was recommended to be considered to improve the analytical prediction accuracy of this phenomenon. This would be attributed to annealing or some kind of damage recovering of the alloy or cyclic loading effect under the identical strain path. Moreover, experimental results show that the pre-strain under plane strain tension had little effect on the shape of FFLCs, which was also observed by the prediction of the DF2012 criterion. The prediction also demonstrated that the pre-strain effect under plane strain tension was negligible for the left-hand side of FFLCs, but slightly lowers the right-hand side FLC, even though the effect was not so obvious. To summarize, the prestrain under plane strain tension had little effect on the formability when the following strain path was between uniaxial tension and plane strain tension, but slightly reduces the ductility of metals in case that the secondary strain path was from plane strain tension to equibiaxial tension. However, the comparison shows that there was a large potential for the improvement of fracture modeling under stretching after plane strain tension. One of the improving approaches was to take the

evolution of yield surfaces or kinematic hardening into account instead of simple isotropic hardening assumption.

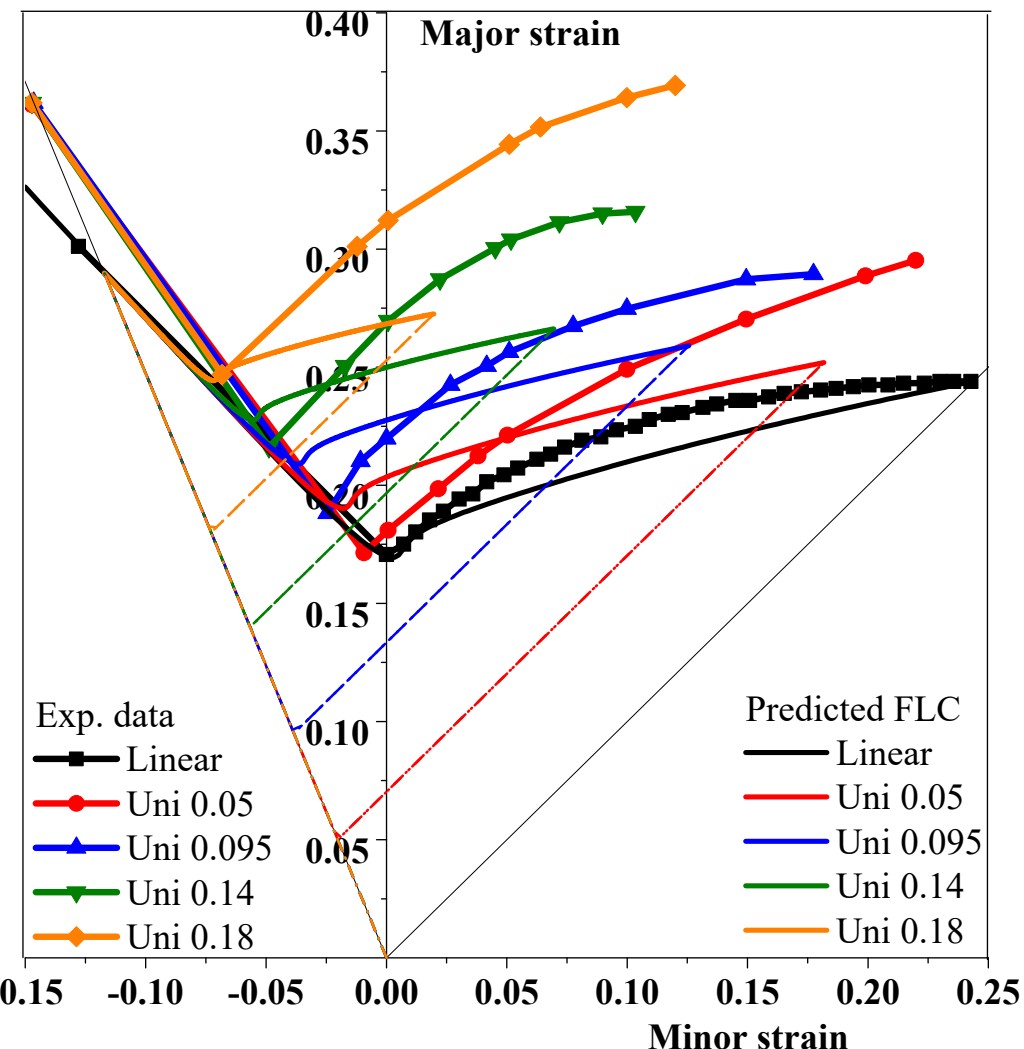

**Figure 4.** FFLC comparison between experiments and prediction by the DF2012 criterion prestrained under uniaxial tension.

The effect of pre-strain under equibiaxial tension was compared between experiments and the DF2012 prediction in Figure 6. Both experiments and prediction show that the pre-strain of equibiaxial tension lowers the formability of metals for most secondary strain paths, except for the case that the secondary strain path was close to uniaxial tension. However, the predicted effect of pre-strain under equibiaxial tension was less than experimental observation, even though a similar effect was predicted by the DF2012 criterion. The experimental results and the DF2012 prediction suggest that the secondary strain paths between plane strain tension and equibiaxial tension after prestrain under equibiaxial tension should be avoided for better formability of sheet metals in the design of forming processed and tool geometry, but the strain path of equibiaxial tension followed by uniaxial tension was preferred since this special loading path improves the deformation limits before fracture. The results also show that the fracture predicting error of equibiaxial tension followed by plane strain tension was very large, and approaches should be proposed to better model the strain path changing effect on fracture limit for this bilinear strain path.

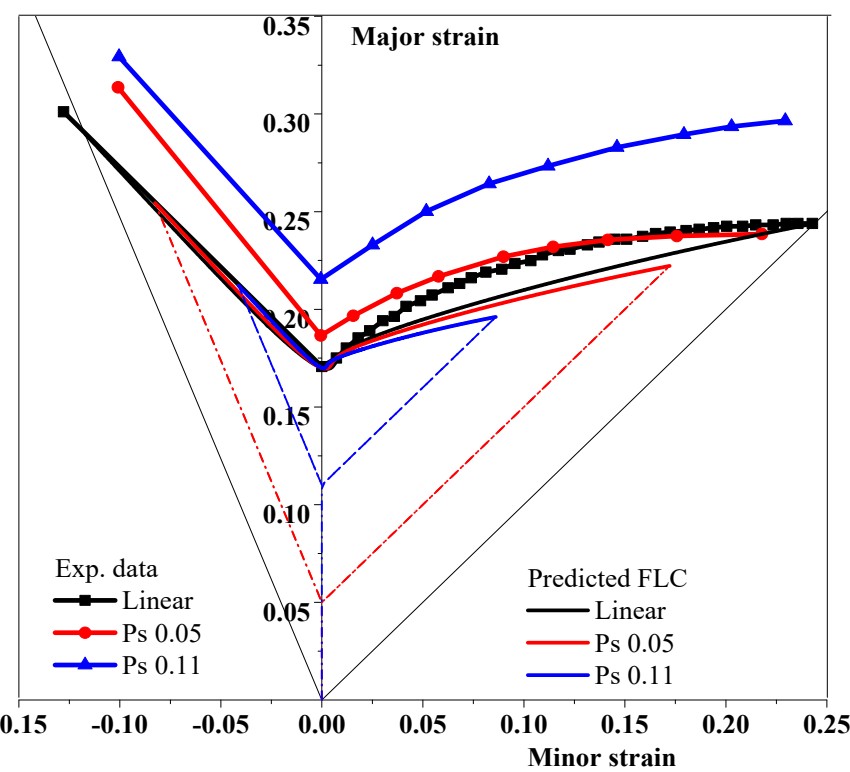

**Figure 5.** FFLC comparison between experiments and prediction by the DF2012 criterion prestrained under plane strain tension.

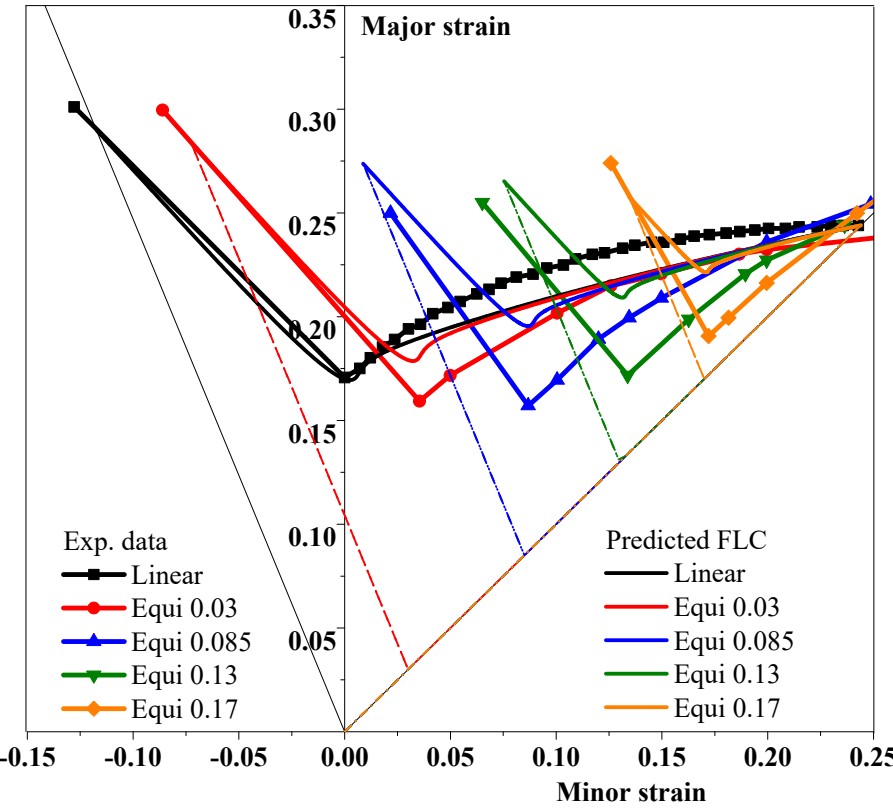

**Figure 6.** FFLC comparison between experiments and prediction by the DF2012 criterion prestrained under equibiaxial tension.

## 5. Conclusions

This study investigates the predictability of the DF2012 criterion for the effect of strain path changing on fracture limits of an aluminum alloy of AA6111-T4. The DF2012 criterion was first shown to be capable of fracture prediction under proportional loading conditions from uniaxial tension to equibiaxial tension. The application of the calibrated DF2012 criterion to FFLC prediction under bilinear strain paths demonstrated that the DF2012 criterion describes the similar effect of pre-strain under uniaxial tension, plane strain tension and equibiaxial tension on FFLCs. However, there was a big difference between prediction and experiments for some cases, including fracture strain of uniaxial tension followed by equibiaxial tension, plane strain tension followed by uniaxial tension, and equibiaxial tension followed by plane strain tension. The reason was probably due to the neglecting of cyclic loading effect in analytical modeling of plasticity. Analytically, predicting accuracy, in this case, would be improved if the evolution of the yield surface or kinematic hardening was considered for plasticity for strain path changes. To summarize, the DF2012 criterion was capable of FFLC modeling accurately under proportional loading and reasonable predicting the strain path changing effect on FFLCs for AA6111-T4. However, the predicting error for some bilinear strain paths still could not be neglected. A better analytical approach should be developed to reduce the difference between prediction and experiments, such as the consideration of kinematic hardening in plasticity modeling for bilinear strain paths.

**Author Contributions:** Conceptualization, S.L. and Y.L.; methodology, S.L. and Y.L.; software, S.L., Y.L. and Y.X.; validation, S.L., Y.L., and G.Y.; formal analysis, S.L. and Y.L.; investigation, S.L., Y.L., G.Y. and Y.X.; resources, S.L. and Y.L.; data curation, S.L. and Y.L.; writing—original draft preparation, S.L. and Y.L.; writing—review and editing, S.L. and Y.L.; visualization, S.L., Y.L. and G.Y.; supervision, S.L. and Y.L.; project administration, S.L. and Y.L.; funding acquisition, S.L. and Y.X. All authors have read and agreed to the published version of the manuscript.

**Funding:** This research was funded by the National Natural Science Foundation of China (grant no. 52075423 and 51905551) and the Taizhou Science and Technology Project of Zhejiang (grant no. 2002gy14).

**Data Availability Statement:** Not applicable.

**Conflicts of Interest:** The authors declare no conflict of interest.

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
