# Peer review of "Prediction of Strain Path Changing Effect on Forming Limits of AA 6111-T4 Based on a Shear Ductile Fracture Criterion"

_metals, doi:10.3390/met11040546_

Round 1

Reviewer 1 Report

According to the review, the manuscript was revised more clearly.

However, English expressions need to be further improved.

Author Response

Thank you very much for your time and efforts. All the authors are sincerely grateful to the reviewer for all your work. We polished our language with our best and all the revisions are highlighted in yellow in the revised manuscript.

Reviewer 2 Report

The paper has been reviewed according to my comments. It should be accepted for publication in Metals.

Author Response

Thank you very much for your time and efforts. All the authors are sincerely grateful to the reviewer for all your work.

Reviewer 3 Report

The article presents a modelling framework, consisting of anisotropic Drucker yield function and ductile fracture criterion, used to obtain forming limit curves for an AA6111-T4 alloy at non-proportional loading paths.

I find the article very lacking in several aspects and do not recommend the publication in the present form. 

The language needs thorough editing. Many sentences are poorly formulated and difficult to understand. Some grammatical errors can also be found.

The experimental data and methods are not reported properly. A single reference to a relatively old article is given with no explanations of the used methods. Practically no material parameters are reported, even the ones used in the calibration of the model. 

Then the authors assume planar isotropy but use an anisotropic yield function. Whether this is supported by the experimental data (not reported) remains unclear. The stress-strain curves or work-hardening law are not reported. The way the non-proportional strain path was applied or how any calculations were made is not reported. 

The results are not analysed well. Especially I found it baffling that the isotropic(? it is mentioned but never reported) hardening model produced different fracture strains for application of the same uniaxial tension strain once or twice. Why an isotropic(?) work hardening would be good at reproducing strain path changes is not clear. It actually does not work so well, but the authors do not attempt to address this and just explain it away with "errors in measurements", or say that "analytical models should be improved". 

Overall it is recommended to rewrite the article properly, without omitting important parts, analyse the results thoroughly and resubmit.

Author Response

Thank you very much for your careful review. The text is fully polished with our best as you suggested in the revised manuscript. All the revisions are highlighted in yellow. Thanks again.

Please refer to the attachment for detailed modification.

Round 2

Reviewer 3 Report

The authors did some changes to the article, but the language was not improved significantly. In addition, they really did not make it clear in the first draft that they did not perform any experiments and took all data directly from Graf and Hosford (1994). The sentence "Experiments were conducted according to Graf and Hosford (1994)" implies the performance of the experiments, which is not the case. 

Turn out that the authors' contribution is applying a ductile fracture criterion on a plasticity function, based on plane stress assumption. The models are applied to non-proportional strain path, even though Lou et al. (2012) do not advise this and propose an integral form of the ductile fracture criterion for such cases. The plasticity model does not have any means to adequately describe the non-proportional strain paths either, like kinematic hardening. As a result, the fit of the model to the experiments is naturally underwhelming. 

Considering all this I do not recommend the publication of the present article, due to the lack of novel and interesting results and the use of oversimplified models not suitable for the given problem.

This manuscript is a resubmission of an earlier submission. The following is a list of the peer review reports and author responses from that submission.

Round 1

Reviewer 1 Report

In the work, a ductile fracture criterion for prediction of fracture forming limit diagrams of sheet metals, presented by the authors themselves in 2012, is calibrated under proportional loading conditions. Then, it is used to predict the response under bi-linear strain paths. The discrepancy between the predictions and the experimental data is shown to discuss the effect of strain path changing on the forming limit curves of the aluminum alloy sheet considered.

The theme is of great interest and the work is clearly presented. However, in my opinion, there are no innovative elements that justify its publication in the paper.

A further point to consider concerns the modeling of the plastic response. For this, an anisotropic model was calibrated. However, in its validation, the response of the model is not compared with the experiments in the different directions, but only the mean value of the Lankford parameter is considered. This does not allow an evaluation of its accuracy which is fundamental for the correct prediction of ductile damage. Therefore, I believe that there are not all the elements necessary to thoroughly evaluate strain path changing effect on forming limits , which is the goal of the work.

Reviewer 2 Report

This paper investigated the effect of the strain path changing on the forming limit curve during deformation using Anisotropic Drucker Yield Function and Shear Ductile Fracture Criterion for aluminum alloy sheet. This paper tried to provide information on the effect of strain paths by comparing and analyzing predictions for various strain paths with experimental data. However, for the following reasons, this paper seems to be problematic in terms of the quality of information provided and the purpose of the paper.
According to the results predicted by the authors, there is a large difference from the experimental data, especially for some strain paths. As the manuscript title strongly implies, the purpose of this paper seems to be to theoretically and analytically 'predict' the effect of strain path changing on FLC. However, in the current manuscript, countermeasures that can reduce the difference from the experiment including modification of the predictive model are not covered. The current manuscript merely shows that there is a difference between prediction and experimental data. If the main purpose of this paper is to explain the effect of strain path on FLC based on experimental data, then the content of experiment should be covered in detail in the manuscript. However, although the experimental results are displayed in FLC, the contents showing the experiment in detail are very insufficient. For this reason, the current manuscript does not seem to provide useful experimental data, nor does it seem to suggest a method to predict FLC taking into account the effects of the strain path.

I recommend that the authors clarify the identity of this paper and revise the manuscript accordingly.

Reviewer 3 Report

This is a very nice paper, interesting for the scientific community. It should be accepted for publication in metals. The authors show that their fracture model DF2012 can predict the FLCs after strain path change, in a good approximation for a pre-strain in tension and in biaxial expansion. Unfortunately the predictions for a pre-strain in plane tension are poor.

Concerning initial assumptions, the authors considered that rbar=0.68 characterises planar isotropy. In my opinion, this parameter characterises normal anisotropy. Planar isotropy is rather measured by Delta r = (r0 - 2r45 + r90)/2 that should be close to 0 for planar isotropy. Please give this value in the paper.

p.5 where sigma bar represent the equivalent stress